# Diagnosing Manifold Collapse: Intervention-Based Topology in Low-Rank RNNs

## Abstract

Understanding how neural circuits give rise to low-dimensional manifolds remains a central challenge in neuroscience and AI. While toroidal and ring-like topologies have been observed, the mechanisms linking connectivity to emergent geometry are not fully understood. We propose an intervention-based topological framework that integrates low-rank recurrent neural networks, persistent homology, and curvature-aware scoring to study the formation, degradation and recovery of neural manifolds. Our analysis compares perturbed and unperturbed trajectories under targeted lesions (random, axis-aligned, low-norm), introducing the Causal Topological Intervention Score (CTIS) to quantify Betti number shifts with curvature weighting. We also develop a Dynamic Betti Fingerprint for anomaly detection and evaluate resilience via the Area Under Recovery Curve (AURC). Using synthetic velocity-driven trajectories and spike-train recordings from the CRCNS pfc-7 dataset, we show that structured low-rank connectivity yields toroidal dynamics, and that CTIS captures non-linear collapse thresholds aligned with recovery trends. This framework offers a principled tool for linking interventions on connectivity to interpretable topological signatures of circuit fragility.

## 1 Introduction

Recent advances in systems neuroscience have established that neural population activity is not randomly distributed in high-dimensional firing spaces but instead evolves on low-dimensional manifolds that encode behaviourally and cognitively relevant variables. These manifolds provide a compact geometric substrate for neural computation, enabling stable, generalisable representations over time. Empirical studies have revealed striking instances of such structure: ring-shaped manifolds in head-direction cells and toroidal topologies in grid cells of the medial entorhinal cortex (Gardner et al., 2022). These observations raise a central mechanistic question:

> *Which features of neural circuit connectivity give rise to specific manifold topologies, and how do these topologies disintegrate under structural perturbations?*

While manifold learning techniques such as Uniform Manifold Approximation and Projection (UMAP) (McInnes et al., 2018) and t-SNE (Van der Maaten & Hinton, 2008) reveal latent geometric organisation, and persistent homology summarises topological structure via Betti numbers, these tools are primarily descriptive. Betti numbers quantify the number of $k$ dimensional holes in a topological space, such as connected components ($\beta_0$), loops ($\beta_1$) and voids ($\beta_2$), and serve as compact descriptors of global structure (Edelsbrunner et al., 2008). However, such methods lack mechanistic explanations for how manifold geometry emerges or deteriorates, and offer no causal insight into how connectivity changes reshape latent dynamics. This limitation is particularly salient in the study of neurodegenerative conditions such as Alzheimer's disease, where progressive synaptic deterioration correlates with disruptions to grid-cell representations and spatial memory. Yet, the mechanistic links remain poorly understood.

Therefore, we introduce an intervention based topological framework that bridges structured neural connectivity with topological signatures in latent dynamics. Our approach centres on low rank recurrent neural networks (RNNs), where the recurrent weight matrix is factorised as $\mathbf{UV}^\top$, with $\mathbf{U}, \mathbf{V} \in \mathbb{R}^{d \times r}$ parameterising a rank-$r$ subspace within the full $d$-dimensional connectivity space. This low rank structure constrains the network's dynamics to a subspace, supporting the emergence

of low dimensional manifolds, an organisation observed in biological systems (Mastrogiuseppe & Ostojic, 2018). By applying biologically motivated perturbations such as random ablations, axis aligned truncations, and pruning of low norm rows, we simulate diverse modes of circuit degradation. To quantify the topological impact, we introduce the *Causal Topological Intervention Score* (CTIS), a novel metric that captures Betti number changes under perturbation, weighted by curvature estimates derived from the spread of principal component analysis (PCA) (Jolliffe & Cadima, 2016) eigenvalues. This weighting reflects the intuition that deformations in high curvature regions are more structurally fragile.

In addition to static analysis, we introduce a *Dynamic Betti Fingerprint* that computes sliding-window persistence to track temporal fluctuations and flag transient anomalies in latent manifold topology. We also model recovery by gradually restoring lesion weights and summarize resilience with *Area Under the Recovery Curve* (AURC). To identify drivers of observed variation in CTIS across conditions, we report SHAP (SHapley Additive exPlanations) (Lundberg & Lee, 2017) attributions over $\Delta$ Betti features, curvature and AURC, and visualize finite-difference sensitivity maps over the recurrent connectivity.

We evaluate our framework on (i) synthetic velocity-driven inputs that generate toroidal dynamics in low-rank RNNs, and (ii) multi-session spike-train recordings from the CRCNS **pfc-7** dataset (prefrontal cortex)[1] (Mizuseki et al., 2014). In the synthetic setting, CTIS increases in geometrically strained regimes and exhibits a non-linear collapse threshold; across synthetic runs, SHAP attributions over scalar features indicate that $\Delta\mathrm{Betti}_2$ typically explains the largest share of *empirical* CTIS variation, consistent with void (toroidal) disruption driving the observed collapse. In the biological setting, Dynamic Betti Fingerprints reveal session-dependent Betti-1 fluctuations; we interpret these as indicators of latent instability rather than claims about toroidal topology.

In summary, our contributions to the community include:

1. An *intervention-based topological* framework linking structured perturbations in recurrent connectivity to *relative* changes in manifold topology (persistent homology).

2. **CTIS**: a *post-hoc*, curvature-weighted metric for quantifying topological degradation under lesions (not used for training).

3. A *Dynamic Betti Fingerprint* for temporal anomaly detection in latent dynamics.

4. An *AURC* protocol to summarize recovery trajectories under gradual weight restoration.

5. *Attribution analyses* (finite-difference sensitivities; SHAP over scalar features) explaining empirical CTIS variation across runs and sessions.

This framework advances the study of neural degradation by enriching geometric and causal approaches with a topology focused perspective grounded in circuit level perturbation analysis.

## 2 RELATED WORK

### 2.1 NEURAL MANIFOLDS AND LOW-DIMENSIONAL DYNAMICS

A substantial body of research has examined how population activity in the brain can be embedded in low-dimensional manifolds. For instance, Gardner et al. (2022) showed that grid cells in the medial entorhinal cortex encode position on a toroidal manifold, supporting continuous attractor models of spatial cognition. On the theoretical front, Langdon et al. (2023) and Mastrogiuseppe & Ostojic (2018) established conditions under which low-rank recurrent neural networks (RNNs) generate structured latent trajectories, including rings and tori. These findings underscore how architectural constraints influence emergent geometric structure, though they primarily offer descriptive insights.

We extend these efforts by examining how such latent manifolds degrade under controlled perturbations. By simulating biologically inspired lesions within low-rank RNNs and assessing topological consequences using persistent homology, we provide a causal perspective linking synaptic changes to alterations in latent manifold structure.

---

[1] https://crcns.org/download

## 2.2 TOPOLOGICAL DATA ANALYSIS IN NEUROSCIENCE

Topological data analysis (TDA) has increasingly been applied to uncover structural features in neural systems. Early work by Dabaghian et al. (2012) demonstrated that hippocampal place cell coactivity encodes topological information about physical space. Subsequent studies, such as Saggar et al. (2022), used topological summaries to monitor brain state transitions in fMRI data, while Vaupel et al. (2023) analysed Betti number fluctuations in population dynamics.

Despite these advances, many TDA applications remain observational, offering descriptive summaries without modelling the causal mechanisms underlying topological change. In contrast, our framework introduces the *Causal Topological Intervention Score* (CTIS), which quantitatively measures the topological impact of structured perturbations in neural circuits. This enables a comparative analysis of intervention effects and supports hypothesis-driven investigations of manifold fragility.

## 2.3 INTERVENTION-BASED CAUSALITY IN DYNAMICAL SYSTEMS

Causal inference has been a longstanding tool in neuroscience for studying temporal interactions, with techniques such as dynamic causal modelling (DCM) (Friston et al., 2003) and Granger causality (Seth et al., 2015) widely employed to estimate directed functional connectivity between neural regions. These methods have provided insight into the dynamics of cortical circuits, particularly in settings involving task-related or stimulus-driven neural responses. Recent advances have proposed more scalable and expressive frameworks for causal discovery in recurrent and nonlinear dynamical systems. For example, (Langdon & Engel, 2025) introduced a differentiable causal discovery approach tailored to latent recurrent neural networks, and (Pamfil et al., 2020) proposed DYNOTEARS, a continuous optimization framework for learning linear causal dependencies in multivariate time series. However, such approaches typically focus on statistical dependencies and do not account for how causal perturbations influence the geometry or topology of the underlying dynamical state space.

Our work addresses this gap by integrating causal reasoning with topological and curvature-aware analysis. The CTIS metric explicitly captures how different lesion types alter the structure of the underlying manifold, thereby connecting causal perturbations with geometric deformation. Our use of "causal" is strictly *intervention-based*: we analyse changes induced by targeted perturbations, without assuming a structural causal model or identification via do-calculus.

## 2.4 GEOMETRY-AWARE REPRESENTATION LEARNING

The role of geometric structure in neural and machine representations has become a growing focus. In neuroscience, representational geometry has been used to characterise cognitive states and transformations (Kriegeskorte & Diedrichsen, 2019). Within machine learning, methods such as curvature-aware message passing and mixed-geometry embeddings have been developed to model hierarchical and non-Euclidean patterns (Sun et al., 2023; Shang et al., 2024). Our framework draws from these perspectives by incorporating curvature sensitivity via principal component anisotropy and modelling structural resilience through manifold recovery metrics.

By considering these elements, low-rank dynamics, topological invariants, causal perturbation modeling and geometric sensitivity, our approach offers a unified framework for understanding how structural changes in recurrent networks manifest in topological space, with implications for both neuroscience and robust model design.

## 3 METHOD

We introduce an intervention–based topological framework designed to dissect how structural perturbations in recurrent neural circuits affect the geometry and topology of latent dynamics. Our approach integrates low-rank recurrent neural networks (RNNs), persistent homology, curvature-aware sensitivity measures, and interpretable attribution tools. It enables a systematic investigation of neural fragility, degradation and recovery, at both global and local topological scales.

### 3.1 LOW-RANK RNN DYNAMICS AND LATENT GEOMETRY

We begin with a low-rank recurrent neural architecture, parameterised by matrices $\mathbf{U}, \mathbf{V} \in \mathbb{R}^{H \times R}$, where $R \ll H$ controls the rank of recurrence:

$$\mathbf{W}_r = \mathbf{U}\mathbf{V}^\top \tag{1}$$

This constraint encourages formation of structured attractor dynamics, such as rings or tori, by limiting the network's capacity and aligning its latent representations along a structured manifold. The recurrent update for hidden states $\mathbf{h}_t \in \mathbb{R}^H$ is given by:

$$\mathbf{h}_{t+1} = \tanh\left(\mathbf{W}_r \mathbf{h}_t + \mathbf{W}_{\text{in}} \mathbf{x}_t\right) \tag{2}$$

where $\mathbf{x}_t \in \mathbb{R}^d$ is the input signal (e.g., velocity vectors or neural activity bins), and $\mathbf{W}_{\text{in}}$ maps inputs into the hidden space. This setting mimics biological circuits where effective connectivity is constrained by low-dimensional anatomical or functional modules.

### 3.2 TOPOLOGICAL SIGNATURE EXTRACTION VIA PERSISTENT HOMOLOGY

To probe the latent geometry, we embed the hidden trajectories $\{\mathbf{h}_t\}$ into a low-dimensional space using UMAP (McInnes et al., 2018). We interpret *relative*, not absolute, topological changes, since the same embedding procedure is applied symmetrically to perturbed and unperturbed trajectories; this mitigates potential distortions from UMAP's global geometry. We then compute persistent homology (PH) using the Ripser library to extract Betti numbers $\text{Betti}_k$, where $k = 0, 1, 2$ correspond to connected components, loops, and voids, respectively:

$$\text{Betti}_k = \#\{k\text{-dimensional topological features persistent across filtration}\} \tag{3}$$

These invariants provide a robust characterisation of the global shape of the manifold and allow us to quantify geometric disruption due to perturbations.

We also compute a *curvature proxy* $\gamma_g \in [0, 1]$, estimated via PCA eigenvalue spread:

$$\gamma_g = 1 - \lambda_1^{\text{PCA}} \tag{4}$$

where $\lambda_1$ is the proportion of variance explained by the first principal component. Larger $\gamma_g$ implies more isotropic latent structure, which often correlates with fragility of topological features.

To capture temporal variations, we define a **dynamic Betti fingerprint** by sliding a window across time:

$$\text{Betti\_Series}_t = \text{PH}(\text{UMAP}(\{\mathbf{h}_t, \ldots, \mathbf{h}_{t+w}\})) \tag{5}$$

This produces a time series of topological signatures, allowing detection of localised collapses, transitions, or instability in latent structure.

### 3.3 CAUSAL TOPOLOGICAL INTERVENTION SCORE (CTIS)

To quantify intervention effects on manifold topology, we define the Causal Topological Intervention Score (CTIS). Our "causal" is strictly *intervention-based*, referring to changes induced by targeted perturbations rather than identification via structural causal models or do-calculus. We define:

$$\text{CTIS} = \gamma_g \cdot \sum_{k=0}^{2} w_k \cdot |\text{Betti}_k^{\text{pre}} - \text{Betti}_k^{\text{post}}| \tag{6}$$

where $\gamma_g$ is the curvature proxy (Sec. 3) and $w_k$ are user-defined weights. We use uniform weights $w_k = 1$ and obtain similar trends under alternative schemes (e.g., $w_2 > w_1 > w_0$). CTIS is a *post-hoc* score that captures the magnitude of topology change, with curvature modulation emphasizing fragile regions.

### 3.4 LESION SIMULATION AND RECOVERY DYNAMICS

To mimic pathological disruption, we introduce structured perturbations into the matrix $\mathbf{U}$:

- **Random lesion:** entries in $\mathbf{U}$ are zeroed independently with uniform probability.
- **Axis-aligned lesion:** entire rows of $\mathbf{U}$ are truncated, simulating loss of neurons or features.
- **Low-norm lesion:** rows with the smallest $\ell_2$-norm are pruned to target weakly active units.

After each lesion, we recompute CTIS and $\Delta\mathrm{Betti}_2$, defined as:

$$\Delta\mathrm{Betti}_2 = \left| \mathrm{Betti}_2^{\mathrm{pre}} - \mathrm{Betti}_2^{\mathrm{post}} \right| \tag{7}$$

We then simulate partial recovery by gradually reintroducing weights from the original $\mathbf{U}$. Let $\mathrm{CTIS}_t$ denote the score after $t$ recovery steps. We define the area under the recovery curve (AURC) as:

$$\mathrm{AURC} = \sum_{t=1}^{T} \mathrm{CTIS}_t \tag{8}$$

which captures the integrative resilience of the system to perturbations. This metric is post hoc and does not affect training, but summarises resilience under weight restoration.

### 3.5 ATTRIBUTION AND ANOMALY DETECTION

We compute finite-difference sensitivity scores to identify sensitive weights in $\mathbf{U}$:

$$S_{ij} \approx \frac{\Delta\mathrm{Betti}_2(\mathbf{U}_{ij} + \epsilon) - \Delta\mathrm{Betti}_2(\mathbf{U}_{ij})}{\epsilon}. \tag{9}$$

These scores are visualised as heatmaps to highlight vulnerable substructures within the network, used as post-hoc diagnostics.

We also apply SHAP (SHapley Additive exPlanations) (Lundberg & Lee, 2017) to regress CTIS from scalar features such as $\Delta\mathrm{Betti}_2$, curvature and AURC. This yields feature attributions that explain *empirical variation* in CTIS across runs and sessions. Finally, we flag anomaly events using a threshold on first-order Betti-1 differences:

$$\mathrm{Anomaly}_t = \mathbb{I}\left( |\mathrm{Betti}_1[t] - \mathrm{Betti}_1[t-1]| > \tau \cdot \sigma \right) \tag{10}$$

where $\tau \in [2,3]$ is a sensitivity hyperparameter and $\sigma$ is the empirical standard deviation of the Betti-1 fluctuations. These events highlight local topological instability in latent dynamics, which we interpret as internal regime shifts.

## 4 EXPERIMENTS

We evaluate our framework on both synthetic and biological neural data to examine how structural perturbations affect latent manifold topology, and how such effects can be detected, quantified, and reversed through topological and intervention-based metrics.

**Synthetic latent dynamics.** We begin with a controlled simulation of 2D velocity trajectories representing continuous spatial movement. These are designed to elicit toroidal dynamics in a low-rank recurrent neural network (RNN). The network, with hidden dimension 64 and rank 4, generates latent trajectories whose topological features are extracted using UMAP and persistent homology. To probe topological robustness, we apply three types of structured perturbations to the recurrent weight matrix: random ablation, axis-aligned truncation, and pruning of low-norm rows. Each intervention produces a modified latent space, which is evaluated via the proposed Causal Topological Intervention Score (CTIS), changes in Betti numbers (with $\Delta\mathrm{Betti}_2$ often most informative empirically), and recovery capability measured by the area under the recovery curve (AURC).

**Real neural recordings.** We further evaluate our framework on the CRCNS pfc-7 dataset (Mizuseki et al., 2014), which provides multi-session extracellular spike recordings from prefrontal cortex during behavioural tasks. This dataset complements our synthetic analysis by capturing naturalistic neural dynamics across multiple sessions, offering a testbed for examining intervention-induced manifold changes. Spike events are binned into 100ms intervals and normalised across recorded units. These time-series vectors are used to drive the same low-rank RNN, enabling analysis of the resulting latent trajectories. We apply identical lesion and recovery protocols as in the synthetic setup and compute topological features using persistent homology and curvature proxies derived from PCA anisotropy.

**Evaluation protocol.**    We assess: (i) the effect of different lesion strategies on CTIS and Betti shifts including $\Delta\text{Betti}_2$, (ii) the dynamics of topological fluctuations via sliding-window Betti series, and (iii) the potential for manifold recovery. We further interpret CTIS through SHAP feature attribution and sensitivity-based heatmaps, linking structural degradation to topological collapse. Experiments across multiple CRCNS sessions show that the framework detects consistent relative topological transitions despite session variability.

**Implementation and code availability.**    All experiments were implemented in Python using Py-Torch, and executed on an NVIDIA RTX 4090 GPU. Each session-level run, including RNN simulation, manifold embedding and topological analysis, is completed in approximately one hour. The framework integrates several established tools: UMAP (McInnes et al., 2018) for nonlinear dimensionality reduction, Ripser (Bauer, 2021) for persistent homology, PCA for curvature estimation, and SHAP (Lundberg & Lee, 2017) for post hoc feature attribution. UMAP is configured with $n_{\text{neighbors}} = 5$ and min_dist $= 0.5$ to prioritise local geometric fidelity (McInnes et al., 2018). An anonymised codebase is included in the supplementary material for review; the full repository with documentation will be released publicly upon publication.

## 5    RESULTS AND DISCUSSION

We evaluate our framework using both synthetic simulations and multi-session biological recordings to analyse how structured perturbations in recurrent connectivity affect the topology of latent neural manifolds. Our framework integrates persistent homology, curvature-aware metrics and attribution analysis to quantify degradation and resilience.

**Synthetic degradation and manifold collapse.**    To assess how structural perturbations impact latent manifold geometry, we simulate two-dimensional velocity-driven trajectories on a toroidal surface and train a low-rank recurrent neural network (RNN) to encode the dynamics. Perturbations are applied by progressively zeroing fractions of the recurrent weight matrix $\mathbf{U}$, modelling controlled synaptic degradation. Figure 1a shows the behaviour of the Causal Topological Intervention Score (CTIS) as lesion severity increases. Instead of rising linearly, CTIS peaks near 40% ablation before declining, indicating that topological collapse is not simply proportional to weight removal. This non-linear trend reflects a critical transition where void structures disintegrate despite preserved partial connectivity. CTIS thus captures fragility in manifold structure that cannot be explained by norm or rank loss alone. Supporting this, Figure 1b visualises the manifold after 50% lesioning. Although geometrically smooth, the toroidal topology is lost, consistent with reductions in Betti-2 and curvature, indicating collapse in higher-order organisation.

These results validate CTIS as a geometrically grounded and biologically plausible diagnostic. Its sensitivity to structured degradation, confirmed both quantitatively and visually, highlights its utility for probing topological resilience in recurrent neural systems.

**Topological irregularities in real neural recordings.**    We assess session-wise latent dynamics in biological data by applying persistent homology to RNN-derived hidden states driven by the CRCNS **pfc-7** spike-train recordings from prefrontal cortex. After UMAP projection into three dimensions, we compute Betti numbers over sliding windows to obtain dynamic topological fingerprints. As shown in Figure 2, Betti-0 remains stable across all time points, indicating persistent manifold connectedness. In contrast, Betti-1 exhibits low-amplitude, transient fluctuations, corresponding to short-lived topological loops in the latent space.

To detect local instability, we apply a threshold of 2.5 standard deviations on the first-order difference of Betti-1. Vertical red lines in Figure 2 highlight automatically identified anomaly points, which are further summarised in the session-wise anomaly raster (Figure 3). These anomalies are temporally sparse and vary across sessions, with Sessions 1 and 8 showing early and widely distributed events. While the amplitude of Betti-1 fluctuations is modest, the presence of abrupt local shifts suggests latent fragility in circuit-level encoding. We interpret these irregularities as indicators of transient instability rather than claims about specific representational transitions.

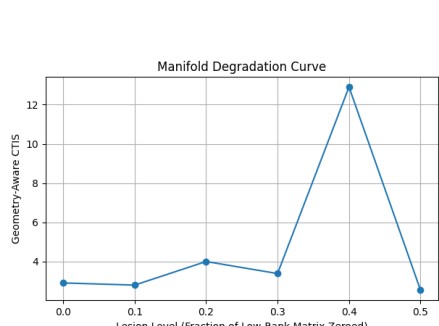
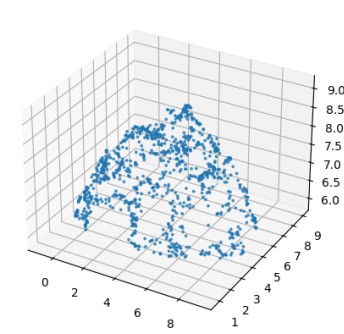

(a) CTIS degradation curve. Peak at 40% lesion indicates topological collapse.

(b) UMAP of hidden states post-lesion. Betti-2 void disappears at 50%.

Figure 1: Synthetic degradation analysis: (a) CTIS curve showing peak collapse, (b) UMAP projection showing void loss after lesion.

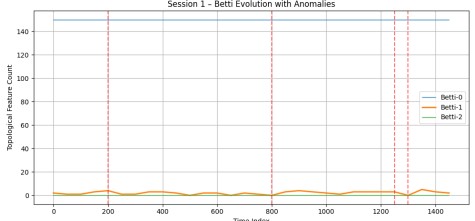
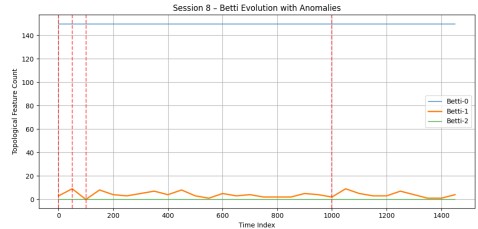

Figure 2: Topological feature evolution for Session 1 (left) and Session 8 (right). Betti-0 remains constant, while Betti-1 exhibits sparse transient fluctuations. Red dashed lines mark detected anomalies based on thresholded first-order differences.

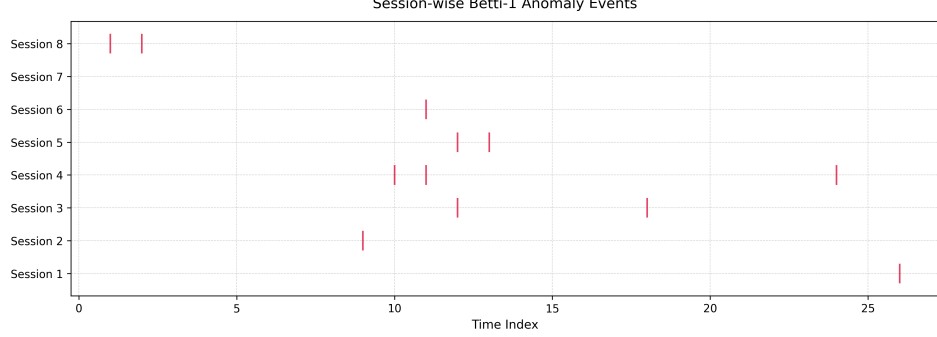

Figure 3: Raster plot of Betti-1 anomaly events across eight sessions. Events are unevenly distributed, with Session 1 and 8 displaying greater temporal spread, suggesting session-specific latent instability.

**Lesion-type analysis: degradation and recovery dynamics.** To examine how different perturbations affect topological degradation and resilience, we stratify results by lesion type: axis-aligned truncation, low-norm pruning, and random ablation. Figure 4a reports mean and standard deviation of CTIS and $\Delta\text{Betti}_2$ across lesion types. Low-norm lesions induce the highest degradation in both metrics, indicating that removing units with small norm values can still strongly affect higher-order

topology. In contrast, axis-aligned perturbations show smaller average degradation, with variability across seeds.

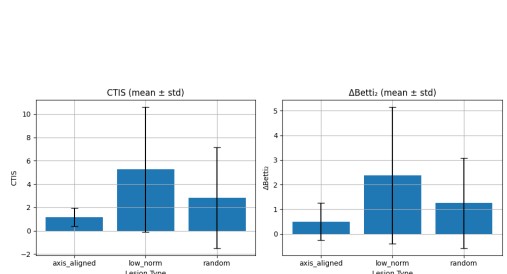
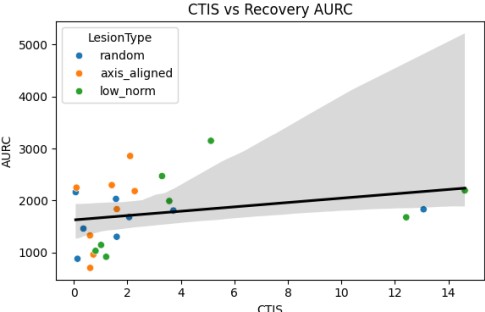

(a) CTIS and $\Delta\text{Betti}_2$ (mean ± std) across lesion types.

(b) CTIS vs AURC: higher CTIS predicts more difficult recovery.

Figure 4: Lesion-type analysis: (a) Topological and geometric disruption across lesion types. (b) Positive correlation between CTIS and recovery difficulty.

We also assess recovery difficulty using the area under the recovery curve (AURC). Figure 4b shows a positive correlation between CTIS and AURC, indicating that severe topological collapse leads to prolonged recovery. Although ANOVA did not reach significance (CTIS: $F = 2.11$, $p = 0.146$; $\Delta\text{Betti}_2$: $F = 1.84$, $p = 0.184$), the effect sizes ($\eta^2 = 0.168, 0.149$) suggest that lesion type accounts for a non-trivial share of variance. To validate this further, we performed Bayesian ANOVA, which revealed that $\Delta\text{Betti}_2$ is highest under *low_norm* lesions (mean = 2.38, 95% HDI: [1.12, 3.84]), followed by *random* (1.29 [0.04, 2.63]), with *axis_aligned* showing lower and uncertain effects (0.50 [–0.90, 1.87]). These posterior estimates reinforce the lesion-dependent nature of topological collapse. CTIS reflects this trend more consistently by integrating curvature and weighting, supporting its value as a geometry-aware diagnostic metric (see further summary in Table 2 in Appendix G). Table 1 consolidates these findings, providing a numerical summary of lesion-specific CTIS, $\Delta\text{Betti}_2$, and AURC values across sessions.

Table 1: Summary of CTIS, $\Delta\text{Betti}_2$, and AURC across lesion types (mean ± std).

| Lesion Type | CTIS | $\Delta\text{Betti}_2$ | AURC |
|---|---|---|---|
| Axis-Aligned | 1.18 ± 0.79 | 0.50 ± 0.76 | 1802.68 ± 740.50 |
| Low-Norm | 5.26 ± 5.34 | 2.38 ± 2.77 | 1822.97 ± 780.99 |
| Random | 2.82 ± 4.32 | 1.25 ± 1.83 | 1645.58 ± 416.61 |

Overall, the metric captures both immediate topological fragility and the downstream challenge of structural repair.

**Attribution analysis and feature sensitivity.** To identify key drivers of intervention severity, We train a random forest regressor to predict CTIS using three interpretable scalar descriptors: $\Delta\text{Betti}_2$, curvature index, and AURC. SHAP analysis (Figure 5) identifies $\Delta\text{Betti}_2$ as the most influential and consistent predictor, with a mean impact of $+2.76$, substantially surpassing curvature ($+0.23$) and AURC ($+0.20$). This supports our hypothesis that the collapse of 2D topological voids, rather than geometric deformation or recovery difficulty, is the dominant signal driving manifold fragility in perturbed dynamics.

To complement the global attributions, we compute finite-difference scores of CTIS with respect to each element of the recurrent weight matrix $\mathbf{U}$. As shown in Figure 6, the resulting heatmap reveals sparse yet structured sensitivity across rank components, indicating that topological collapse is mediated by a limited subset of critical connectivity directions. These patterns reinforce CTIS as a Betti-aware fragility score responsive to specific perturbations in latent dynamics.

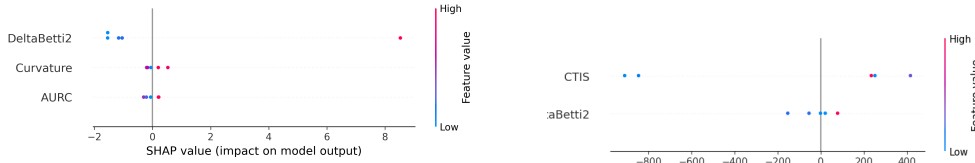

Figure 5: SHAP-based attribution for predicting CTIS. Left: global mean absolute impact across features. Right: beeswarm summary plot showing per-sample SHAP values and input magnitudes. $\Delta\mathrm{Betti}_2$ dominates across both global and local attribution.

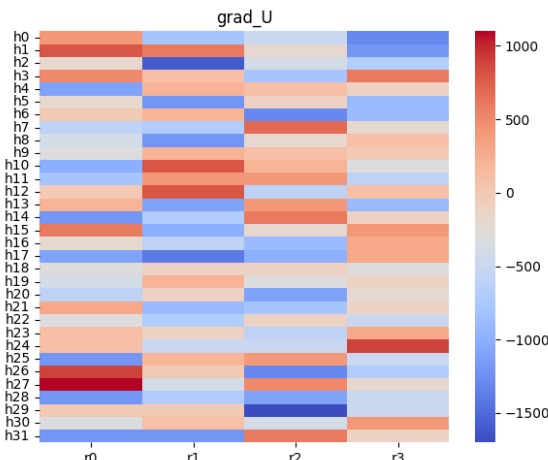

Figure 6: Sensitivity of CTIS with respect to recurrent weights $\mathbf{U}$. Each cell shows $\frac{\partial \mathrm{CTIS}}{\partial \mathbf{U}_{ij}}$ via finite differences. Structured anisotropy indicates a sparse set of perturbation-sensitive axes.

## 6    CONCLUSION

We introduce an intervention-based topological framework to analyse how structured perturbations to recurrent connectivity shape latent dynamics. Using low-rank RNNs with persistent homology, curvature-aware metrics and SHAP attribution, we define the Causal Topological Intervention Score (CTIS), which couples curvature and Betti differences to quantify structural fragility. Experiments show a non-monotonic CTIS that peaks before full degradation, indicating that topological collapse is not linearly related to the extent of weight removal. Although classical ANOVA does not yield group-level significance, consistent effect sizes and Bayesian posteriors across lesion types indicate that CTIS is more sensitive to lesion-specific disruption than raw Betti metrics. SHAP pinpoints Betti-2 collapse as the main driver of CTIS variation, finite-difference scores reveal sparse yet structured sensitivity in the recurrent weights, and real recordings exhibit session-specific latent instability.

Our framework provides a framework for diagnosing topological degradation, but several limitations remain. Curvature is currently estimated via linear PCA and separating embedding-induced effects from genuine session dynamics remains challenging. Future work will incorporate higher-order topological fingerprints, formal causal discovery on persistence diagrams, and learned recovery dynamics. We also anticipate applications in AI systems, where measures of topological resilience could serve as diagnostics for structural generalisation failure.

## 7    REPRODUCIBILITY STATEMENT

For reproducibility during review, anonymised source codes, results and datasets are included in the supplementary material. More details are also included in the appendix D, E and G.

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

## A    ETHICS STATEMENT

We use only public data sets `https://crcns.org/data-sets/pfc/pfc-7/`; no new data was collected and no personally identifiable information is known. Experiments are for research use, and more details are reported to support reproducibility in the appendix D and E.

## B    USE OF LARGE LANGUAGE MODELS

Large Language Models were used to help draft and polish text only. All technical content, experiments, and analyses were designed, executed and verified by the authors.

## C    IMPACT STATEMENT

This work introduces a intervention based topological framework for analysing how structural perturbations in neural connectivity affect the geometry and topology of latent dynamics, with implications for both biological and artificial systems. By formalising the Causal Topological Intervention Score (CTIS) and dynamic Betti fingerprinting, the framework offers a principled diagnostic perspective on how recurrent circuits degrade and recover under structured disruption. This is particularly relevant for spatially organised systems such as the medial entorhinal cortex (MEC), where manifold topology is often used as a proxy for circuit-level stability. We emphasise that our claims are computational and intervention-based rather than clinical.

Notably, the ability to detect early collapses in Betti-2 features and quantify recovery dynamics parallels key signatures of synaptic degradation seen in neurodegenerative conditions. Consequently, the proposed methodology may support early-stage detection and progression tracking in computational models of disease, highlighting potential clinical applications in Alzheimer's monitoring. These findings suggest broader relevance for neuroscience and healthcare.

Beyond biological contexts, our approach advances interpretability in machine learning by linking low-rank connectivity, topological invariants, and causal perturbations. Potential applications include assessing resilience in recurrent neural networks, modelling degradation in neural simulations,

and guiding robust AI system design through geometry-aware diagnostics. By considering causal and topological reasoning, the framework offers a novel lens for understanding and safeguarding structure-sensitive neural systems.

## D  DATASET ACCESS AND PREPROCESSING

All real neural data used in this study are sourced from the publicly available CRCNS pfc-7 dataset `https://crcns.org/download`. The dataset comprises multi-session extracellular recordings from prefrontal cortex during rodent navigation. It can be accessed via:

- **URL**: `https://crcns.org/data-sets/pfc/pfc-7/`
- **Registration**: A free CRCNS.org account is required to download the dataset.

In our experiments, spike trains are binned using a fixed bin size of 100 ms and normalised by z-scoring across channels. Only well-isolated units are retained. Processed spike matrices are then passed into our low-rank RNN model for manifold and topological analysis. Details of the binning pipeline and preprocessing script are included in the anonymised supplementary materials and will be made publicly available on GitHub upon acceptance.

## E  TECHNICAL IMPLEMENTATION DETAILS

The full pipeline is implemented in Python using PyTorch for model definition and training, shown in Figure 7, and Scikit-learn, UMAP-learn and Ripser.py for manifold projection and persistent homology.

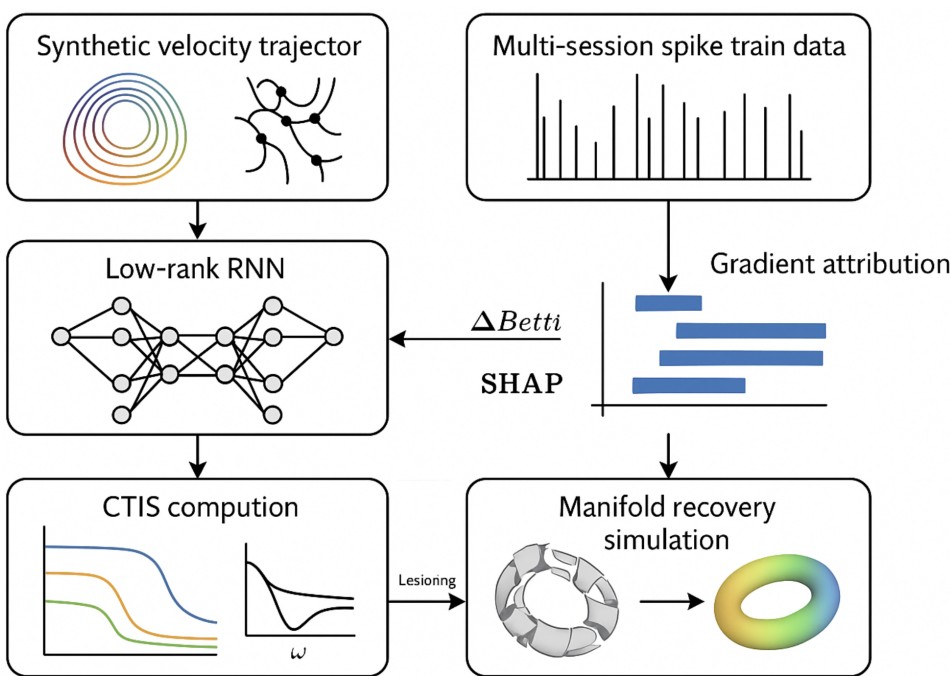

Figure 7: Overall work flow of the proposed method

- **Model**: The recurrent neural network is low-rank constrained with hidden dimension 64 and rank 4. Input dimensions are determined by the number of recording channels in each session.
- **Activation function**: `tanh`.
- **Initialisation**: Weight matrices $\mathbf{U}$ and $\mathbf{V}$ are initialised with Gaussian noise.

- **Persistent homology**: Computed using Ripser with maxdim $= 2$ and filtration threshold $= 0.8$.

- **Curvature index**: Estimated from PCA eigenvalue anisotropy over local neighbourhoods.

- **CTIS**: Calculated using fixed weights $(1, 1, 1)$ across Betti-0, Betti-1 and Betti-2 shifts, scaled by curvature. We treat CTIS as a post-hoc score for comparative analysis.

- **Sliding window for topological dynamics**: 600 timesteps with a stride of 50.

- **SHAP attribution**: Performed using a random forest regressor with 100 estimators, trained to predict CTIS based on scalar inputs: $\Delta\text{Betti}_2$, curvature, and AURC.

- **Sensitivity attribution**: Finite-difference scores of $\Delta\text{Betti}_2$ with respect to elements in $\mathbf{U}$ are computed to identify locally sensitive directions of perturbation.

All experiments were run on a single Nvidia RTX 4090 GPU. Average runtime per session (including training, topological analysis and attribution) is approximately one hour. The complete codebase and reproducibility scripts are provided as anonymised supplementary material and will be made publicly available via GitHub upon acceptance.

## F    METHODOLOGICAL CLARIFICATIONS AND CLINIC RELEVANCE

This section supplements the main text with technical justifications, biological context, and evidence-based rationale for core modelling choices. The emphasis is on interpretability, reproducibility, and clinical plausibility, particularly in the context of Alzheimer's-related degradation.

**Curvature Index and Fragility Sensitivity.**    We define the curvature index as $\gamma_g = 1 - \lambda_1$, where $\lambda_1$ is the first PCA eigenvalue of the local embedding. While this is an approximation that captures local anisotropy rather than formal Riemannian curvature, prior work has shown that such PCA-based surrogates correlate with fragility in synthetic manifolds (Bubenik et al., 2020). In our experiments, increases in $\lambda_1$ ("flattening") tend to co-occur with Betti-2 loss and higher CTIS. This aligns with reports of medial entorhinal cortex (MEC) instability in early Alzheimer's disease, where disruptions in grid-cell structure indicate latent geometric degradation (Zhao et al., 2022). We treat $\gamma_g$ as a neuroscience-motivated heuristic for weighting potentially fragile regions, not a validated biomarker.

**Low-Rank RNNs as Structural Probes.**    Instead of analysing task-optimized RNN models, we employ low-rank RNNs to isolate how the network's intrinsic architecture (specified by a factorized recurrent weight $\mathbf{W}_r = \mathbf{U}\mathbf{V}^\top$) shapes its latent dynamics. This approach parallels prior studies of neural dynamics that examine spontaneously generated cortical activity and the emergence of low-dimensional attractor states in random networks (Sussillo & Abbott, 2009; Mastrogiuseppe & Ostojic, 2018). Thus, our framework offers a tractable testbed for introducing controlled architectural perturbations and conducting fragility analysis on the resulting dynamics, all without the confounding influence of any learning signals.

**Biological Plausibility of Lesion Types.**    Our model's lesion strategies are designed to mirror canonical neurodegenerative damage patterns. Random ablation (zeroing out a random subset of weights) mimics the widespread synaptic loss characteristic of early Alzheimer's disease (Shankar & Walsh, 2009). Axis-aligned pruning (removing an entire neuron by deleting a row of weight matrix $U$) approximates a focal brain lesion, akin to a microinfarct, a tiny stroke that destroys a localized cluster of synapses (Smith et al., 2012). Low-norm pruning (eliminating the neuron with the smallest total weight magnitude) targets the network's least active unit, mirroring how amyloid-$\beta$ pathology triggers activity-dependent synaptic weakening and loss via long-term depression (Koffie et al., 2011). This biologically grounded spectrum of lesion types supports an interpretable analysis of the RNN's resilience under various damage motifs.

**Clinical Outlook.**    The observed phenomena, Betti-2 collapse, curvature flattening and delayed recovery, mirror disruption patterns identified in medial entorhinal cortex (MEC) circuits affected by Alzheimer's disease. Studies have shown that early Alzheimer's pathology disrupts grid-cell periodicity and destabilizes the MEC's topological encoding of space. For instance, tau pathology induces excitatory neuron loss, grid cell dysfunction, and spatial memory deficits reminiscent of

early Alzheimer's disease (Zhao et al., 2022; Fu et al., 2017). Our dynamic Betti tracking and CTIS metrics offer a mechanistic, interpretable complement to black-box classifiers, capturing circuit-level fragility through topological and curvature-aware fingerprints. This framework may support early detection or intervention strategies by linking structural degradation in latent dynamics to underlying synaptic vulnerability.

## G   BAYESIAN ANOVA TEST RESULT

Table 2 presents posterior estimates from the Bayesian ANOVA model assessing the effect of lesion type on $\Delta\text{Betti}_2$. The *low_norm* condition shows the highest mean and a 97% HDI that excludes zero, indicating strong evidence for increased topological collapse. The *random* condition shows moderate evidence, while *axis_aligned* has a lower mean with wider uncertainty overlapping zero. The posterior for $\sigma$ reflects moderate residual variability. These results reinforce lesion-type-specific effects observed in the main analysis.

Table 2: Posterior estimates of $\Delta\text{Betti}_2$ by lesion type. Means and 95% Highest Density Intervals (HDI) are reported.

| Lesion Type | Mean | SD | HDI 3% | HDI 97% |
| --- | --- | --- | --- | --- |
| axis_aligned | 0.500 | 0.742 | -0.898 | 1.874 |
| low_norm | 2.382 | 0.738 | 1.119 | 3.840 |
| random | 1.290 | 0.699 | 0.041 | 2.627 |
| $\sigma$ | 2.042 | 0.325 | 1.497 | 2.665 |

These posterior intervals are reported for transparency, that are exploratory rather than conclusive, given session-level variability.

