# OpenReview forum: "Diagnosing Manifold Collapse: Intervention-Based Topology in Low-Rank RNNs"
_ICLR.cc/2026/Conference — ICLR 2026 Conference Withdrawn Submission_

### Official Review · Reviewer_Rujq · 2025-10-31

**Soundness:** 3
**Presentation:** 4
**Contribution:** 2
**Rating:** 4
**Confidence:** 4

**Summary:**

Summary

The manuscript proposes a novel framework to study the formation of
neural manifolds and their degradation and recovery following
perturbations. The aim is to establish causal links between changes in
connectivity and subsequent changes in the topology of neural
manifolds.



Soundness

Most of the manuscript is sound. The authors build their metrics on
top of well-known topological features (Betti numbers). SHAP is
applied to gain explainability insights and to identify the most
influential measures contributing to variations in CTIS. Low-rank
networks are suitable for this analysis and provide a good testbed for
linking connectivity, dynamics, and topology. The authors also explain
the metrics in intuitive terms while providing detailed justifications
in the appendix.

A puzzling point is the authors' claim that they are able to capture
the curvature of the manifold. A related question arises from this
(see questions).

In addition, the authors mention in the Introduction and Results &
Discussion sections that the low-rank recurrent network is trained,
yet no details are given about the type of training or its effects on
the read-in or recurrent weights that produce the toroidal dynamics.


Presentation

The presentation of the work is good and provides good explanations
that can be followed intuitively, also by readers who are not experts
in the mathematics of topology.


Contribution

The main contribution is the proposal of metrics and their causal
relationship to manifold integrity and collapse.  This framework
allows the study of topological breakdown in a controlled model
setting using either artificial or biological input data.

**Strengths:**

I very much like the authors' style of presentation, providing the
main equations and clear intuitive explanations of the different
metrics.  The manuscript carefully dissects the causal effects of
perturbations on the various metrics.  The authors draw thoughtful
links to, and distinctions from, previously established measures.

**Weaknesses:**

PCA-based method for measuring curvature requires additional
description and clarification in the main text.  The training
procedure of the recurrent neural networks (RNNs) is not described.
If I understand correctly, Figure 3 should summarize all sessions,
including sessions 1 and 8 shown in Figure 2 above, but the detected
anomaly events do not seem to match.

**Questions:**

My concern on the PCA method be in able to capture the curvature of the manifold (see "soundness").

At least based on the global PCA eigenvalue spread described in Section 3.2, this does not appear to be the case.
While we do not doubt that the PCA spectrum may correlate with the fragility or stability of the manifold, PCA is a linear method that only captures the Gaussian variance structure of the data.
It effectively approximates the data by an n-dimensional hyperplane rather than a curved manifold given the first n eigenvalues are retained.
Therefore, the proposed curvature proxy in Eq. (4) seems to reflect dimensionality rather than curvature per se.

In the methodological clarifications, the authors mention a 'local embedding' rather than the global data.
We acknowledge that if PCA is performed locally and its variation across the manifold is analyzed, such an approach could indeed capture curvature-related effects.
We therefore suggest that the authors clarify precisely what they mean by 'local PCA' and briefly explain why this can be interpreted as a measure of curvature in the main text.
Otherwise, the current description could be misleading.

It would be interesting to see whether SHAP would then ascribe more importance to the curvature.

In case these points can be clarified in the rebuttal I am happy to raise my score.

---

> ### Author Response · Authors · 2025-11-30
>
> We thank you for the encouraging overall assessment and for highlighting key clarifications.
>
> &nbsp;
>
> **Clarification on the PCA-based curvature measure**
> We are not estimating Riemannian curvature but rather a global anisotropy index of the point cloud. We will state in Section 3.2 that γg = 1 - λ1 is a global anisotropy (or “dimensionality dispersion”) measure computed on the entire embedded manifold, where λ1 is the leading eigenvalue of the PCA covariance. High γg means variance is distributed across many principal directions (more isotropic, ball-like), whereas low γg​ indicates concentration along a single direction (more anisotropic, line-like).
>
> As discussed in our response to another reviewer, we hypothesise that isotropic manifolds are more globally fragile.
> For local embedding, in the current work, PCA is applied globally to the UMAP embedding, not locally. The suggestion to move towards local PCA as a more faithful curvature approximation is excellent and will be explicitly listed as a direction for future work.
> Finally, given that γg​ is a global heuristic, its relatively low SHAP importance (Fig. 5) compared with the direct topological measure ΔBetti2 is unsurprising and, in fact, reassuring. It indicates that while the anisotropy weighting adds a useful geometric nuance, the dominant contribution to CTIS comes from the raw topological change.
>
> &nbsp;
>
> **Training procedure of the RNNs**
> We appreciate the opportunity to clarify this point. The RNNs are not trained on an explicit task via gradient descent. As described in Section 4 and Appendix F, we use low-rank RNNs as structural probes: a randomly initialised low-rank network is driven by the input (synthetic velocity or real neural data), and its intrinsic low-rank dynamics generate the latent trajectories that we analyse. There is no task loss and no parameter learning. We will restate this in Section 3.1 to avoid the impression that we are performing conventional supervised training.
>
> &nbsp;
>
> **Discrepancy between Fig. 2 and Fig. 3**
> Thank you for catching this inconsistency. The anomaly events in Fig. 3 should indeed align exactly with the red lines in Fig. 2 for the corresponding sessions. This appears to be an error in figure generation or labelling. We will correct this in the final version and ensure that the two figures are consistent.

---

### Official Review · Reviewer_69M8 · 2025-10-31

**Soundness:** 1
**Presentation:** 2
**Contribution:** 1
**Rating:** 2
**Confidence:** 4

**Summary:**

The authors study the effects of destructive interventions to a low-rank RNN on a variety of topological features, analogizing their study to the degredations to qualitative neuronal activity that occur during degenerative disease. They first define a series of topological (and other) signatures to be measured from their RNN: a residual variance after PCA on trajectories, Beti numbers measured over sliding windows, and differences in Betti numbers before and after interventions. These interventions involve zeroing-out weights in the RNN at a given time. The signatures are then measured on the interventions on synthetic toroidal dynamics and action potential measurements from prefrontal cortex.

**Strengths:**

1. I think the question the authors address is interesting and timely. From what I can tell, there is not much work on detecting or measuring changes in topological structure through persistent homology after perturbations. In that sense, the contribution is conceptually novel.

2. The authors provide several different metrics for detecting topological changes, which on the one hand is nice as it should provide a fuller picture of these changes, though as I will mention below also make the approach seem scattered.

3. I appreciate the effort to use real neural data, and I think this shows that the proposed metrics can--at least in principle--be applied to this data. This lays the groundwork for future work to investigate topological changes in neural activity.

**Weaknesses:**

1. The paper feels overloaded with metrics and lacking in deeper analysis. I count around five scores (Betti_Series_t, CTIS, AURC after recovery, Betti sensitivity, Anomaly_t) that are sometimes interconnected. I appreciate the approach to be thorough, but given the weakness of the empirical results overall, I believe it would have been better to focus one or another metric to be explored in greater depth. As it stands, I find it rather hard to read and remember each score, which are only very briefly introduced. Also, some scores do not make much sense to me. The explained variance gap $\gamma_g$ is reported as "[correlating] with the fragility of topological features." This is vague, and not immediately intuitive. A long thing (i.e. anisotropic) structure with low $\gamma_g$ could also be argued to be topologically "fragile", no? One small bump and you're out of the structure, perhaps changing the topology. Without clearer explanation, it's hard to know if my intuition is right or wrong. It's further unclear why these values should be used in weighting CTIS. On this quantity, are pre/post measured on adjacent ablations (e.g. .1 vs .2 and .2 vs .3 in Fig 1a) or only between no ablations and stronger ablations (e.g. 0 vs .1 and 0 v .2 and 0 vs .3, etc.)

2. Overall, the empirical results are unfortunately not very strong. I list some areas of weakness figure by figure.

Fig. 1a. The peak at .4 probably represents something, but it's not clear immediately what this is. Again, can you clarify what CTIS measures? Between no ablation and increasingly strong ablations or between neighboring ablation values? If the former, it's not clear why the CTIS wouldn't permanently jump to ~12. And why not go all the way to full ablation?

Fig. 1b. This is very hard to interpret without seeing the manifold before. Can you do a side-by-side of all of the manifolds under ablation, ideally highlighting where you think the structure disappears?

Fig. 2a. The "sparse transient fluctuations" are not easy to see, both because the effect is seemingly small and the figures are not formatted for easy viewing. If Betti 1 is the important one and the other Betti numbers have different scales, then simply show Betti 1. Further, it is hard to know what to think of these detected anomalies. If you have no ground truth or comparison between subjects/conditions, how are we to know that these fluctuations are not just a fluke of your thresholding? Ideally we'd see, say, in condition 1 you have the fluctuations but in condition 2, you don't. Could you simulate this on your synthetic toroidal data?

Fig. 3. My previous results on anomaly detection hold here. If there is no systematic difference between sessions related to ground truth signals or labels, then it is not clear what these anomalies mean. Put another way, maybe you just always get some small fluctuations in noisy data that gets picked up as changes in Betti 1? True, there is work saying that PH is stable to small amounts of noise, but you need to argue in detail why you believe your noise is small and these anomalies are, therefore, indicative of a real topological signal.

Fig. 4a. It is hard to know how to compare the effects of the different lesion types since they are not balanced by magnitude. For example, maybe a low_norm ablation is more destructive in some absolute sense than an axis_aligned ablation, so the changes in CTIS are superficial. Is there any way to control for this?

Fig. 4b. As the authors say, the results are not significant. The Bayesian ANOVA shows that changes in Betti numbers vary differently with lesion type, but what is the significance of this if recovery time is not significantly related to CTIS? Also, the main text says that CTIS has value as a "geometry-aware diagnostic metric", but aren't we talking about topology? This is an important terminological distinction I believe.

Fig. 5,6. I appreciate the attempt to interpret the results with SHAP (which should be explained more in the main text for unfamiliar readers like myself) and sensitivity measures, but my point still stands that if CTIS is not clearly measuring anything significant, then it is not worth explaining or interpreting.


3. I find the figures hard to read. Could axis labels for smaller figures be increased?

**Questions:**

1. I think the paper would start to get a good footing if the authors provided one simple, intuitive example which shows how a lesion is clearly detected by CTIS (or whichever metric) where it *should* be. By *should*, I mean that it would be on an example where we can see by eye that the topology has changed in the Betti sense. So, my first question is, could the authors provide this example, ideally by visualizing the change in topology in an obvious way and showing CTIS peaks at that moment? I believe this is what the synthetic experiments intended to be, but it is not convincing for the moment. I would be willing to start bumping my scores up if I saw this clean toy example.

2. Can you explain your reasoning about $\gamma_g$? I don't see the intuition about topological fragility and isotropy.

---

> ### Author Response · Authors · 2025-11-30
>
> Thank you for the detailed engagement with the metric design and empirical sections. We appreciate the opportunity to clarify the conceptual hierarchy and tighten interpretation.
>
> &nbsp;
>
>
> **Metrics**
> We agree that the set of metrics can feel crowded. Our intention is to provide complementary views:
>
> -*CTIS as a measure of global fragility*
>
> -*Dynamic Betti fingerprints as a view of temporal instability*
>
> -*AURC as a notion of resilience over time*
>
> -*SHAP/sensitivity maps for attribution*
>
> An isotropic manifold γg distributes variance across many directions and is, in our hypothesis, more globally fragile to diffuse perturbations. We compute γg once on the pre-lesion manifold and then use it to weight the post-hoc Betti differences.
>
> &nbsp;
>
>
> **Empirical results**
>
> -*Fig. 1a/b.* CTIS is computed between the intact manifold and each progressively lesioned manifold. CTIS peaks at the point where the Betti-2 void collapses; after that, further lesioning cannot reduce Betti-2 any further, so the metric becomes dominated by smaller Betti-0 and Betti-1 changes, leading to the observed drop.
>
>
> -*Figs. 2/3.* We agree that, in the absence of a ground-truth condition, the biological interpretation of Betti-1 anomalies must remain cautious. In the current Results section we already state: “We interpret these irregularities as indicators of latent instability rather than claims about specific representational transitions.” We will emphasise that these plots primarily serve as a proof-of-concept for detecting transient irregularities, not as definitive claims about neural coding.
>
>
> -*Fig. 4a.* The perturbations are constructed to be comparable in terms of the number of parameters ablated (e.g. 20% of weights or rows). The result that low-norm pruning is most destructive under this constraint is therefore not trivial; it suggests that weights with small magnitude can still be crucial for preserving global topology.
>
>
> -*Fig 4b.* We acknowledge that the frequentist ANOVA is not significant. However, the Bayesian ANOVA (Table 2, Appendix G) indicates that lesion type has an effect on ΔBetti2​. The positive correlation in Fig. 4b between CTIS and AURC suggests that larger initial topological damage is associated with slower recovery. We will clarify that CTIS is termed “geometry-aware” because it combines topological information (Betti numbers) with a geometric weighting γg.
>
>
> We agree that a single, unambiguous example is crucial, and Fig.1 is intended to play that role. We will explicitly annotate Fig. 1a to indicate that the CTIS peak coincides with the lesion fraction at which the Betti-2 void collapses. Our goal is for a reader to see how CTIS tracks the topological collapse.
>
> &nbsp;
>
>
> **Figure readability**
>
> We agree that the figures can be improved typographically. We will increase axis label sizes, annotation clarity, and marker sizes throughout to improve readability.

---

### Official Review · Reviewer_DB5B · 2025-11-01

**Soundness:** 2
**Presentation:** 2
**Contribution:** 2
**Rating:** 2
**Confidence:** 3

**Summary:**

This paper proposes an intervention-based topological framework for studying how recurrent connectivity shapes and disrupts low-dimensional neural manifolds. The authors use low-rank RNNs as model systems and apply a combination of UMAP (for embedding), persistent homology (for Betti-number extraction), and curvature-weighted metrics to quantify manifold collapse under different perturbation types. The central metric, the Causal Topological Intervention Score (CTIS), summarizes changes in Betti numbers and curvature across “lesions” in network weights. Experiments are performed on both synthetic velocity-driven trajectories (toroidal dynamics) and spike-train recordings (CRCNS pfc-7).

While the paper raises an interesting question—how structural perturbations relate to changes in representational topology—the proposed framework appears highly empirical and under-validated. The strong reliance on UMAP, lack of sensitivity and baseline comparisons, and limited statistical support make it difficult to assess either robustness or novelty in a convincing way.

The paper explores an intriguing question and presents a creative combination of geometric and topological tools. However, the empirical results are not sufficiently rigorous to support the main claims. The dependence on UMAP, absence of sensitivity and baseline analyses, lack of statistical support, and limited connection to concrete neuroscience or ML use cases collectively weaken the contribution. With stronger validation, clearer visual explanation, and a more principled discussion of the embedding choice, this line of work could become more impactful.

**Strengths:**

* Interesting conceptual framing: The “intervention-based” idea of connecting network perturbations to topological changes is creative and potentially useful for both computational neuroscience and model diagnostics.
* Brings together multiple perspectives: The integration of low-rank RNNs, topology (persistent homology), and causal perturbations is conceptually appealing, and could serve as a prototype for connecting geometry and causality in neural dynamics.
* Potential cross-disciplinary relevance: The questions about “manifold/circuit fragility” and recovery could be meaningful if extended to empirical neuroscience or robust ML systems.

**Weaknesses:**

* Overreliance on UMAP without justification: The entire analysis depends on UMAP embeddings, yet the paper neither explains why UMAP is appropriate nor evaluates whether the results are stable under different embeddings or hyperparameters. Given that UMAP is known to be sensitive to parameters such as n_neighbors and min_dist, this is a serious limitation. At the very least, the paper should test robustness empirically or justify the choice theoretically.
* Lack of statistical rigor: Several figures (e.g., Fig. 1A, Fig. 4) lack error bars or show wide overlaps, making trends ambiguous. For instance, the sudden CTIS drop between 0.4–0.5 lesion fraction in Fig. 1A is unexplained. Without uncertainty quantification or multiple runs, the evidence for the claimed non-linear “collapse threshold” is weak.
* No baseline or ablation comparisons: The paper introduces several post-hoc metrics (CTIS, AURC, curvature weighting) but does not compare them against simpler alternatives (e.g., using raw persistent homology, PCA instead of UMAP, or unweighted Betti differences). As a result, it is unclear what value CTIS adds.
* Weak connection to neuroscience or ML relevance: While the “intervention” framing is promising, the work does not clearly link its findings to real neuroscientific hypotheses or machine-learning implications. It reads more as an exploratory RNN case study than a framework addressing a specific gap.
* Limited interpretability and missing illustrations: Given the geometric and topological focus, the paper should include schematic diagrams showing how manifold structures deform or collapse under lesions. The absence of visual intuition makes the framework hard to follow.
* Writing clarity: The exposition is serviceable but somewhat descriptive; motivation and framing could be tightened to emphasize the research question and the intended scope of generalization.

**Questions:**

1. How robust are the results to UMAP parameters (n_neighbors, min_dist) or to alternative embeddings (e.g., PCA, Isomap)?
2. Could you provide a clear interpretation of the sharp CTIS drop between 0.4–0.5 lesion severity in Fig. 1A?
3. Have you tested whether your results hold without curvature weighting or without UMAP (i.e., using direct persistent homology on high-dimensional trajectories)?
4. What specific neuroscientific or ML problems could CTIS help diagnose beyond this toy low-rank RNN setup?
5. Can you include schematic figures to illustrate intuitively how manifold collapse is detected and what CTIS represents?

---

> ### Author Response · Authors · 2025-11-30
>
> Thank you for the careful review and helpful suggestions. We summarise the key clarifications as follows.
>
> &nbsp;
>
>
> **UMAP justification**
> We agree that the dependence on UMAP needs to be better motivated. Our choice was guided by the need to preserve both local and global manifold structure more faithfully than linear methods such as PCA, which is important for subsequent persistent homology.
>
> In Section 3.2, we already state that “We interpret relative, not absolute, topological changes, since the same embedding procedure is applied symmetrically to perturbed and unperturbed trajectories; this mitigates potential distortions from UMAP’s global geometry.” The analysis focuses on relative changes in topology (∆Betti) between pre- and post-lesion states under an identical embedding pipeline, which reduces sensitivity to UMAP-specific global distortions.
>
> As detailed in Appendix E, we use n_neighbors = 5 and min_dist = 0.5 to prioritise local geometric fidelity, which is a standard setting in manifold learning. We will add explicit justification in Section 3.2.
>
> &nbsp;
>
>
> **Statistical clarity**
> For Fig. 1a, the sharp CTIS drop between lesion fractions 0.4–0.5 corresponds to a topological phase transition where the toroidal void (Betti-2) collapses. This is not intended to represent a gradual degradation but a critical threshold at which the manifold’s structure changes qualitatively, as visualised in Fig. 1b. We will make this interpretation explicit in the figure caption.
>
> You are correct that Fig. 1a currently lacks uncertainty quantification. The synthetic result shown is a single representative run chosen to illustrate the mechanism clearly. In the revised version, we will include variability across five random seeds (e.g., as shaded regions) to show that the non-monotonic CTIS behaviour is robust.
>
> For the biological multi-session data, we already report variance (e.g., Table 1, Fig. 4) and use Bayesian ANOVA (Appendix G) to provide probabilistic effect size estimates. We will make clearer in the main text that we deliberately favour a Bayesian treatment over p-value-based testing, given the exploratory nature of the study.
>
> &nbsp;
>
>
> **Baselines and ablations**
> The curvature proxy γg is a heuristic weighting that emphasises topological changes occurring in more isotropic (and hence, in our view, more fragile) regions of the manifold. We will also clarify that PCA, as a linear method, fails to recover the non-linear toroidal structure in the first place, making the intended topological analysis infeasible. The choice of UMAP is therefore not arbitrary but driven by the need to uncover the relevant manifold structure.
>
> &nbsp;
>
>
> **Neuroscience and ML relevance**
> We appreciate the opportunity to clarify this.
>
> -*Neuroscience.*
>  The work is motivated by the degradation of structured representational manifolds in neurodegenerative disease, which we mention in the abstract and Introduction. We model several biologically inspired lesion types (random, axis-aligned, low-norm) and relate the observed manifold degradation to grid-cell disruption in Alzheimer’s disease, as discussed in Appendix F. We will more clearly signpost this connection in the main text.
>
>
> -*Machine Learning relevance.*
> We see CTIS as a diagnostic tool for “topological degradation” and “structural generalisation failure” in learned representations. In particular, it provides a way to characterise how recurrent architectures lose structured manifold geometry under perturbations. We will refine the conclusion to make this role more concrete.

---

### Author Response · Authors · 2025-11-30
**Review and Reviewer-Author Discussion Summary (1/2)**

Dear PCs, SACs, ACs, and Reviewers,
&nbsp;

Thank you very much for your thoughtful reviews and constructive suggestions. To assist the AC, we summarise below the key strengths highlighted by the reviewers and how we addressed their main concerns during the discussion.

**Strength**
Overall, we are grateful that the reviewers recognised substantial conceptual and methodological strengths in the paper. Specifically:

-**Novel intervention-based framing of topology under perturbations.**
The paper proposes an intervention-based topological framework to study how structural perturbations affect neural manifolds, using CTIS and related metrics to link lesions to manifold collapse. **All three reviewers recognised this point** (DB5B: Strengths 1–3, 69M8: Strength 1, Rujq: Contribution).





-**Integration of low-rank RNNs, persistent homology, and causal perturbations.**
The framework combines low-rank RNN dynamics, UMAP-based embeddings, Betti numbers, and lesion interventions to study manifold integrity in a controlled setting. This integration was explicitly noted as appealing and potentially cross-disciplinary (DB5B: Strengths 2–3, Rujq: Soundness & Contribution).


-**Multiple complementary metrics for manifold degradation.**
The use of CTIS, dynamic Betti time series, AURC, and sensitivity/SHAP analyses was seen as providing a richer view of topological change, despite concerns about overload. This was positively noted by Reviewer 69M8 (Strength 2) and by Reviewer Rujq (Soundness).


-**Clear presentation and intuitive explanations.**
Reviewers appreciated the combination of main equations with intuitive explanations of the metrics and causal effects of perturbations. Reviewer Rujq explicitly praised the presentation (Presentation: 4, Strengths), and DB5B highlighted the clarity of intuition behind the metrics.


-**Use of real neural data as proof of applicability.**
The application to CRCNS prefrontal cortex data was recognised as an important step showing that the proposed metrics can, in principle, be used beyond synthetic RNN dynamics. Reviewer 69M8 explicitly acknowledged this (Strength 3).

---

> ### Author Response · Authors · 2025-12-01
> **Review and Reviewer-Author Discussion Summary (2/2)**
>
> **Concerns and Our Addressing.**
> During the discussion, we focused on clarifying key methodological and interpretative points. The main concerns and our responses are summarised below.
>
> -**Embedding choice and dependence on UMAP**
>  (DB5B: Weakness “Overreliance on UMAP”; 69M8: Question on robustness and PCA/Isomap)
>
>
> **Our Addressing**
>
> We clarified that UMAP is used because PCA cannot recover the non-linear toroidal manifold that underpins the persistent homology analysis. We emphasised that all conclusions are based on relative topology (ΔBetti) computed symmetrically for intact and lesioned trajectories under the same embedding pipeline, which mitigates sensitivity to UMAP’s global geometry. We also made our parameter choices (e.g. neighbors=5 , min_dist=0.5 and rationale explicit in Section 3.2.
>
> -**Statistical clarity and interpretation of CTIS (especially Fig. 1)**
>  (DB5B: Weakness “Lack of statistical rigor”; 69M8: Questions on Fig. 1a peak and definition of CTIS)
>
> **Our Addressing.**
>
> We clarified that CTIS is computed between the intact manifold and each lesioned manifold, and that the peak around lesion 0.4–0.5 reflects the collapse of the Betti-2 toroidal void; beyond this point, Betti-2 cannot decrease further, so CTIS is dominated by smaller Betti-0/1 changes, explaining the drop. We also pointed out that variability is already reported for biological data via Bayesian ANOVA (Appendix G) and made this connection clearer in the main text. For synthetic data, we explained that the current figure shows a representative run chosen to illustrate the mechanism.
>
> -**Metric overload, hierarchy, and the curvature proxy γg**
>  (69M8: Weakness “overloaded with metrics”; Question on anisotropy/intuition; Rujq: Weakness on curvature interpretation)
>
> **Our Addressing.**
>
> We provided a clearer hierarchy: CTIS for global fragility, dynamic Betti fingerprints for temporal instability, AURC for recovery dynamics, and SHAP/sensitivity maps for attribution. We clarified that γg ​=1−λ1​ (with λ1the top PCA eigenvalue) is a global anisotropy index, not a curvature estimate, and is computed once on the pre-lesion manifold to weight subsequent Betti changes. We acknowledged that PCA is linear and that γg​ reflects dimensionality/anisotropy rather than Riemannian curvature; we listed local PCA–based curvature estimation as future work. This directly addresses the concerns from both 69M8 and Rujq.
>
> -**Neuroscience / ML relevance and interpretation of anomalies**
> (DB5B: Weakness “weak connection”; 69M8: Concerns on Figs. 2–3 and meaning of anomalies)
>
> **Our Addressing.**
> We clarified that the work is motivated by degradation of structured manifolds in neurodegenerative disease, and that the lesion types (random, axis-aligned, low-norm) are chosen to echo plausible patterns of connectivity disruption, with links to grid-cell degradation discussed in Appendix F. On the ML side, we framed CTIS as a diagnostic tool for “topological degradation” and “structural generalisation failure” in recurrent representations. For anomaly detection in biological data, we explicitly tempered the claims, stating that Betti-1 irregularities are interpreted as indicators of latent instability rather than definitive neural transitions, and that these plots serve as proof-of-concept demonstrations.
>
> -**Training procedure for low-rank RNNs** (Rujq: Weakness on missing training details)
>
> **Our Addressing.**
> We clarified that the low-rank RNNs are not trained on a task via gradient descent; instead, they serve as structural probes. Randomly initialised low-rank networks are driven by input (synthetic velocity or neural data), and their intrinsic dynamics generate the latent trajectories analysed. We explicitly restated this in the methodology to avoid the impression of conventional supervised training.
>
> -**Figure readability and consistency**
>  (69M8: Readability issues; DB5B: request for schematic; Rujq: mismatch between Figs. 2 and 3)
>
> **Our Addressing.**
> We acknowledged the readability issues and indicated that axis labels, annotations, and marker sizes will be increased. We also identified the mismatch between Fig. 2 and Fig. 3 as a plotting/labeling error and committed to correcting it so that anomaly events align consistently. Finally, we agreed to add a schematic-style illustration to make the notion of manifold collapse and CTIS more intuitive.
>
> **Recognition of our clarifications from reviewers.**
>
> Reviewer Rujq explicitly stated a willingness to raise their score if the curvature and training clarifications were addressed, and Reviewer 69M8 indicated that a clearer, intuitive example and explanation would substantially improve their assessment. Our rebuttal directly targets these points without altering the core claims or framework.
>
> Above, we have faithfully summarised the reviewers’ comments and our responses. We are grateful to the reviewers, AC, SAC, and PCs for their time and constructive feedback!
>
>
> Sincerely,
> &nbsp;
>
> Authors

---

### Note · Authors · 2026-01-28

I have read and agree with the venue's withdrawal policy on behalf of myself and my co-authors.

---

### Meta-Review · Area_Chair_7zXK · 2026-01-08

**Summary:**

The paper introduces an intervention-based framework to study formation, collapse, and recovery of low-dimensional neural manifolds in low-rank RNNs using UMAP embeddings and persistent homology; The paper proposes CTIS, Dynamic Betti Fingerprints, and AURC as diagnostic metrics, with demonstrations on synthetic toroidal dynamics and CRCNS neural data. The rebuttal addresses several clarity issues- motivating UMAP, explaining CTIS behavior at collapse thresholds, organizing the metric hierarchy, correcting the “curvature” claim to a global anisotropy heuristic, clarifying that RNNs are not trained, and committing to figure fixes - but leaves main main concerns unresolved. In particular, the method remains highly dependent on a single embedding pipeline without convincing robustness or sensitivity analysis;  statistical support for synthetic results is still weak, baseline and ablation comparisons are limited, and the biological anomaly analyses remain proof-of-concept without ground-truth validation. Consequently, in my assessment, despite improved exposition, the key concerns about rigor, robustness, and evidentiary strength remain, and the review scores clearly support a reject decision.

**Reviewer Concerns:**

Regarding robustness to embedding choice / UMAP sensitivity (DB5B), there is no convincing empirical sensitivity or alternative-embedding study. Furthermore, regarding baselines and ablations (DB5B, 69M8), comparisons are limited, and in terms of biological anomaly interpretation, the latter remains proof-of-concept without ground truth or strong controls. Finally, I think even after the rebuttal, the paper falls short of clearly demonstrating added value of CTIS beyond an incremental benefit over simpler measures.

**Reviewer Scores:**

Basically, the "Reject" scores are very unlikely to change. Core concerns on UMAP dependence, weak statistics, and missing baselines remain insufficiently resolved. The same is true for major objections about metric overload, weak empirical signal, and lack of ground truth. Even though I think the "Marginally below ..." score might go up (due to a good rebuttal in terms of the raised issues), the paper is still not ready for publication at ICLR.

---

### Decision · Program_Chairs · 2026-01-26

Reject